# Bayesian neural networks with Dirichlet-process priors on the data-generating-distribution for randomized value-functions based deep reinforcement learning

## Abstract

We introduce a new class of Bayesian Neural Networks (BNNs) which capture (Bayesian) uncertainty in predictions by exploiting the uncertainty about the underlying training-data-generation-distribution via treating it as a random variable distributed according to Bayesian nonparametric priors on the space of distribution functions, i.e. Dirichlet Processes (DPs). We show that these DP based BNNs provide a generalized Bayesian framework for designing randomized value-function based deep reinforcement learning (RL) algorithms. Crucially, RL with DP-BNNs enables to introduce a "prior" mechanism in a principled Bayesian manner. In the past, such a "prior" mechanism has been shown to be decisive (Osband et al., 2018) in the success of randomized-value function based deep-RL algorithms, and a principled Bayesian procedure remained unknown.

## 1 Introduction

A fundamental aspect of Reinforcement Learning (RL) is the exploration-exploitation dilemma: in order to maximize cumulative reward, agents need to trade-off what is expected to be best at the moment, (i.e., exploitation), with potentially sub-optimal exploratory actions. Solving this trade-off in an efficient manner to maximize cumulative reward is a significant challenge as it requires uncertainty estimates. Furthermore, exploratory actions should be coordinated throughout the entire decision making process, known as deep exploration, rather than performed independently at each state.

*Thompson Sampling* (Thompson, 1933; Russo et al., 2018; Riquelme et al., 2018) provide an elegant approach that tackles the exploration-exploitation dilemma by utilizing uncertainty estimates via maintaining a posterior over models and choosing actions in proportion to the probability that they are optimal. In general these models are taken to a be a parametric class of distributions, and priors and posteriors defined over the parameter of that class (e.g. sufficient statistic of an exponential family), however, nothing stops one to use other models in principle, and perform Thompson sampling with those models if it is feasible to sample from their posterior-distributions Riquelme et al. (2018). One natural choice of these statistical models is Bayesian counterpart of Neural Networks (BNNs), given their rich universal function approximation and rich representation learning property. In contrast to a vanilla NN, a BNN elicits a prior distribution over neural network parameters $\theta$. Given $n$ training observations $z_{1:n}$, one estimates a posterior distribution over $\theta$, which in turn captures uncertainty in predictions.

In the context of deep RL, one can represent the *value functions* with such a Bayesian neural network (BNN), and perform efficient exploration based on uncertainty estimates over value-function predictions through a Thompson sampling heuristic . However, a caveat is that maintaining posterior distributions over Bayesian neural networks is intractable for all but the simplest models, and significant effort has been dedicated to approximate Bayesian methods for deep neural networks. These range from Monte-Carlo (MC) dropout (Gal & Ghahramani, 2016) , variational methods (Blundell et al., 2015) to stochastic minibatch Markov Chain Monte Carlo (Neal, 2012) among others. Unfortunately, deep-RL based on these approximately Bayesian methods do not perform well because of the inefficiency of these approximate methods to capture epistemic-uncertainty in the value function estimates (Osband et al., 2018) which is necessary for efficient exploration.

Largely, this inefficiency can be owed to the challenge of designing meaningful priors over the neural network parameter space (Arbel et al., 2023), major reason being that in general it is easier to know some prior information about the generating distribution of the training data (the state, action, reward sequences in RL for example), and to transform that into corresponding (often product) parametric priors on a large number of parameters $\theta$ is not straightforward.

To overcome these limitations, we propose a new class of BNNs that utilizes the uncertainty about the underlying generative distribution of the data through Dirichlet Process (DP) priors for estimating epistemic uncertainty in predictions. Notably, the uncertainty quantification through DPs can be delineated into a data-driven (Bayesian Bootstrap) and an *artificial-history*/"prior" component (Vashishtha, 2026; Vashishtha & Maillard, 2025), and this strength is transferred correspondingly to the randomized value function RL algorithms based on DP based BNNs. In other words, DP-BNNs enable design of randomized value function based RL algorithms that can combine the strength of Bootstrap based algorithms (Osband et al., 2016) with a principled (Bayesian nonparametric) mechanism for generating-artificial-history/incorporating-prior-knowledge. This latter aspect has been shown to be vital in the success of randomized value function based algorithms for *deep-exploration* (Osband & Van Roy, 2015; Osband et al., 2018). We summarize our contributions as follows,

- We introduce Bayesian Neural Networks (BNNs) with Dirichlet Process (DP) priors, a class of BNNs that capture uncertainty in predictions by exploiting the uncertainty about the underlying training-data-generation-distribution via treating it as a random variable distributed according to Bayesian nonparametric priors on the space of distribution functions, i.e. DPs, and we utilize stick breaking representation of DPs to implement DP-BNNs to compare their performance with other Bayesian procedures on a toy data-set. More comprehensively, standard BNNs operate by putting parametric priors on the weights of the neural networks, and then utilizing different approximation schemes, e.g. dropout, to sample from the very high-dimensional, complicated posterior over the neural-network models. These approximately Bayesian methods are not effective in appropriately capturing the epistemic uncertainty in predictions necessary for efficient exploration in reinforcement learning (Osband et al., 2018)

- We show that DP-BNNs provide a generalized framework for randomized value function based deep RL algorithms and we contribute a version of DQN based on DP-BNNs and exhibit its success in a challenging *deep-exploration* benchmark. Specifically, when value functions are represented using DP-BNNs, they capture uncertainty in value function estimates through two mechanisms: first by randomizing the state, action, reward trajectories in the replay buffer through (Bayesian) Bootstrap, and second by introducing additional uncertainty through a proper Bayesian mechanism for artificial-history/"randomized priors" (Osband et al., 2018) through the base measure of the DP priors

We begin by giving necessary background on Dirichlet Processes in Section 3, then discussing in Section 4 the DP based BNNs along with a numerical example on a toy dataset comparing their performance with other Bayesian procedures. This will lead us to Section 5 where after some background on reinforcement learning, we introduce a deep Q learning algorithm with value functions represented by the DP-BNNs and exploration performed through Thompson sampling, instead of the $\epsilon$-greedy, and we also exhibit performance of this version of DQN on a challenging deep-exploration benchmark. We conclude in Section 6.

## 2 Related works

The closest line of works to ours concerns deep RL with randomized value functions wherein, instead of using a vanilla neural network for evaluating point estimates of value functions and relying on epsilon-greedy type of exploration schemes, one performs principled exploration based on Thompson sampling (Thompson, 1933) with neural network posteriors for value-functions. For Thompson sampling with neural networks, most natural method would be to utilize Bayesian neural networks that represent epistemic uncertainty via approximating the posterior distribution over parameters of a base neural network (Neal, 2012). However, a

challenge is the computational cost of posterior inference, which becomes intractable even for small networks (MacKay, 2003), and even approximate inference with MCMC becomes prohibitive for large scale models (Welling & Teh, 2011). Other methods such as Monte-Carlo dropout (Gal & Ghahramani, 2016) and variational-inference (Blundell et al., 2015) fall short of capturing epistemic uncertainty properly enough for efficient exploration in deep RL (Osband et al., 2018). A popular (non-Bayesian) approach has been that of approximating the posterior distribution through (nonparametric) Bootstrap-ensemble (Osband et al., 2016) of neural networks, and their variants with additive randomized prior networks (Osband et al., 2018) (for deep exploration benchmarks) . However, the computational cost in this class of methods significantly increases e.g., the bootstrapped-ensemble incurs a computation overhead that is linear in the number of bootstrap models. Another approach utilizes Bayesian linear regression in the last layer of the neural network (Azizzadenesheli et al., 2018) for approximating the posterior distribution of value-functions, however the utility of this Bayesian parametric approach for deep-exploration benchmarks (e.g. deep-sea) remains unclear. In contrast to above works, DP based approach in this paper is a principled Bayesian nonparametric approach for estimating epistemic uncertainty and, thanks to its Bayesian nature, does not require ensemble of neural networks for approximating posterior samples from value function networks, and only entails an additional constant cost (per-episode) of generating artificial history, beyond the cost of vanilla DQN (Mnih et al., 2015), and is capable of deep-exploration.

## 3 Background on Dirichlet processes

Before discussing the main algorithm proposed in this paper, it is important to concretely discuss a few key aspects concerning Dirichlet Processes, and this is what we do in this section.

**Dirichlet distribution** is a multivariate generalization of the Beta distributions. We denote the Dirichlet distribution of parameters $(\alpha_1, ..., \alpha_n)$ by $\mathtt{Dir}(\alpha_1, ..., \alpha_n)$ whose density function is given by $\frac{\Gamma(\sum_{i=1}^{n} \alpha^i)}{\prod_{i=1}^{n} \Gamma(\alpha^i)} \prod_{i=1}^{n} w_i^{\alpha^i - 1}$ for $(w_1, ..., w_n) \in [0, 1]^n$ such that $\sum_{i=1}^{n} w_i = 1$

**Dirichlet Processes** In the Bayesian formalism, an unknown object is treated as a random variable which is then assumed to be drawn from a prior distribution. A Bayesian solution requires developing methods of computation of the posterior distribution from this prior based on available information about the unknown object. When the unknown object is a probability measure (a cumulative distribution function in the present paper, to be precise), one then faces a non-trivial question of how to even define a prior on an infinite dimensional object and also take care of the constraints of a probability measure (sum up to 1 over its support). An elegant solution was offered in Ferguson (1973) wherein the author introduced the idea of a Dirichlet process (DP) – a probability distribution on the space of probability measures which induces finite-dimensional Dirichlet distributions when the data are grouped. To look at it concretely, consider a random probability measure, $G$, on some nice (e.g. Polish) space $\Theta$ (e.g. $\mathbb{R}$). $G$ is said to be DP distributed with base probability measure $F$ (e.g. a Gaussian, Beta, Bernoulli, etc) and concentration parameter $\alpha \in \mathbb{R}^+$, denoted as $G \sim \mathtt{DP}(\alpha, F)$, if

$$(G(A_1), ..., G(A_r)) \sim \mathtt{Dir}(\alpha F(A_1), ..., \alpha F(A_r))$$

for every finite measurable partition $A_1, ..., A_r$ of the space $\Theta$.

Having witnessed the construction of DP priors on the space of probability measures, one naturally wonders, how to derive posteriors from these priors, and for that we discuss the important property of *conjugacy* in some nonparametric priors.

**Conjugacy** In the Bayesian parametric framework, one can usually use Bayes rule for deriving posteriors for parametric models, however for non-parametric case, Bayes rule cannot be used in general. Posteriors for some nonparametric priors can be derived utilizing the property of conjugacy. Particularly, an observation model $M \in \mathcal{G}$, and the family of priors $\mathcal{Q}$ are called conjugate if, for any sample size $n$ and any observation sequence $X_1, ..., X_n$, the posterior under any prior $Q \in \mathcal{Q}$ is again an element of $\mathcal{Q}$. Also, merely possessing the property of conjugacy is not enough to form a viable Bayesian prior. For example, a generalization of

DPs is the so-called Neutral To The Right (NTTR) processes (Lijoi et al., 2010), while the entire family of NTTR is known to be conjugate, but besides the specific case of DPs, there's no known explicit method of obtaining *posterior indices* for any other member of the NTTR family. This leads us to discuss the form of DP posteriors next.

**Dirichlet Process posteriors**   Let $X_1, \ldots, X_n$ be a sample from an unknown real-valued distribution $G_0$ where $X_i \in \mathbb{R}$. To estimate $G_0$ from a Bayesian perspective we put a prior on the set of all distributions $\mathcal{G}$ and then we compute the posterior distribution of $G_0$, given $\mathbf{X}_n = (X_1, ..., X_n)$. Let's put a DP prior on the set $\mathcal{G}$. Correspondingly, Let $\texttt{DP}(\alpha, F_0)$, denote the DP prior. The distribution $F_0$ can be thought of as a prior guess at the true distribution $G_0$. The number $\alpha$ controls how tightly concentrated the prior is around $F_0$. With a DP prior on $G_0$, the posterior of $G_0$, given $\mathbf{X}_n = (X_1, ..., X_n)$, enjoys *conjugacy*, i.e, it is itself a DP given as, $\texttt{DP}(\alpha_n, \overline{F}_n)$, where, the *posterior indices*, $\alpha_n$, and $\overline{F}_n$ are obtained as follows (Ferguson, 1973; Ghosal, 2010),

$$\alpha_n = \alpha + n, \ \overline{F}_n = \frac{n}{\alpha + n} F_n + \frac{\alpha}{\alpha + n} F_0 \tag{1}$$

Here $F_n$ is the *empirical distribution function* given $X_1, ..., X_n$, i.e., $F_n(x) = \frac{1}{n} \sum_{i=1}^{n} \mathbb{I}(X_i \leq x)$.

Note how the posterior index, $\overline{F}_n$, exhibited in Equation 1 combines the information from observations (via the empirical cdf, $F_n(x)$ ) with that available from the prior (using $F_0$). This is a crucial property of DPs that our algorithm , DPPS, shall harness in order to account for information obtained via observed data, and the prior information. One can easily see that as $\alpha \to 0$, DPs can only account for uncertainty obtained via observations, with no role of prior anymore, and we discuss this next.

**Bayesian Bootstrap**   A very useful idea in statistical inference has been that of Statistical Bootstrap (Efron & Tibshirani, 1994), and a Bayesian version of Bootstrap was introduced in  (Rubin, 1981). Interestingly, this Bayesian version of Bootstrap can also be derived as a special case of the DP posteriors (Ghosal & van der Vaart, 2017). Specifically, the weak limit, $\texttt{DP}(n, \sum_{i=1}^{n} \delta_{X_i})$, (also referred to as the *noninformed limit* sometimes) of the DP posterior, $\texttt{DP}(\alpha_n, \overline{F}_n)$, as $|\alpha| \to 0$ is known as Bayesian Bootstrap (BB), and is given as,

$$\text{BB}_n := \texttt{DP}(n, \sum_{i=1}^{n} \delta_{X_i}) = \sum_{i=1}^{n} W_i \delta_{X_i} \tag{2}$$

where $\mathbf{W}_n = (W_1, ..., W_n) \sim \texttt{Dir}(1, ..., 1)$, and $X_i$ are the observed data points. The mean of a random distribution drawn from Bayesian-Bootstrap can be easily seen to be the dot-product between the weights and the observed data-points, i.e.,

$$\mu(BB_n) = \sum_{i=1}^{n} W_i X_i = \langle \mathbf{W}_n, \mathbf{X}_n \rangle \tag{3}$$

Next we discuss an important representation of DP priors/posteriors that make them amenable to practical applications.

**Stick-breaking representation of DPs**   With the necessary details about DP prior and posterior distributions set, one naturally asks how to draw sample from these distributions because this is necessary if one wants to do any sort of statistical inference using DPs. Particularly, the form of DP posterior (indices) in Equation1 provide little information to answer this question. A representation of random measures sampled from DPs, reported in  (Sethuraman, 1994), known as Stick Breaking representation of DPs, provides an answer to this question. In general, Stick-breaking measures (Ishwaran & James, 2001) are almost surely discrete random probability measures that can be represented as,

$$Q(\cdot) = \sum_{i=1}^{N} q_i \delta_{Z_i}(\cdot) \tag{4}$$

where $\delta_{Z_i}$ is a discrete measure concentrated at $Z_i$, and $q_i$ are random weights, generated independent of $Z_i$, such that $q_i \in [0, 1]$, and $\sum_{i=1}^{N} q_i = 1$. As one can guess, this is analogous to breaking an actual stick into pieces, and hence the name. In a seminal paper, Sethuraman (1994) reported that if these weights, $q_i$, are constructed such that,

$$q_1 = V_1, \ (q_i)_{i=2}^{N-1} = V_i \prod_{j=1}^{i-1}(1 - V_j), \ q_N = \prod_{i=1}^{N}(1 - V_i) \tag{5}$$

$$V_i \overset{iid}{\sim} \text{Beta}(1, \alpha), \ Z_i \overset{iid}{\sim} F, \ i = 1, 2, ...N \tag{6}$$

and $N$ is $\infty$, then the generated random discrete measure, $Q$, in Equation 4 (with $N$ as $\infty$) is such that, $P \sim \text{DP}(\alpha, F)$. Of course, for computation one can't have $N$ as $\infty$, and the infinite series is truncated at some finite $N$, such that a probability mass, $q_N = 1 - \sum_{i=1}^{N-1} q_i = \prod_{i=1}^{N}(1 - V_i)$, is put at the last point, $Z_N$, and this construction ensures that all weights, $q_i$ sum up to one. This finite Stick-breaking representation has been widely used (Ishwaran & James, 2001; Muliere & Tardella, 1998) thanks to its provable optimality in closely approximating the infinite series (see also Appendix Section A for this and for more details on choosing finite $N$, etc).

**Iterative form of DP posterior**  With the stick-breaking representation of DP priors at hand, one wonders how to sample from DP posteriors (i.e. $\text{DP}(\alpha_n, \overline{F}_n)$) in a practically feasible way and, for this, an iterative form of DP posterior (Blackwell & MacQueen, 1973; Sethuraman, 1994) comes in handy, and is given as follows,

$$Q_i(\cdot) \overset{d}{=} V_i \delta_{X_{i-1}} + (1 - V_i)Q_{i-1}(\cdot) \tag{7}$$

Here $V_i \sim \text{Beta}(1, \alpha + i)$, and $\overset{d}{=}$ denotes equality in distribution. Beginning with a random measure sampled from the DP prior, $Q_0$, generated using the stick-breaking method (Equations 5–6), the recursion in Equation 7 is simply applied $N$ times (corresponding to $N$ observations $\{X_1, ..., X_N\}$) to obtain a random measure, $Q_N$, sampled from the DP posterior (i.e. $\text{DP}(\alpha_n, \overline{F}_n)$),

$$Q_N \overset{d}{=} V_N \delta_{X_N} + \sum_{i=1}^{N-1}\left[V_i \prod_{j=i+1}^{N}(1 - V_j)\right]\delta_{X_i} + \left[\prod_{i=1}^{N}(1 - V_i)\right]Q_0. \tag{8}$$

## 4  Bayesian neural networks with DP priors

In this section, we present a new nonparametric methodology for sampling neural-network models from their posteriors and correspondingly capturing (Bayesian) uncertainty in their predictions. The core underlying idea is that uncertainty in the predictions stems from limited knowledge about the true underlying distribution of the data. This can be understood from a simple example of a neural network with parameters $\theta$ and loss $L_\theta(\cdot)$. If one knows the distribution. $G^*(\cdot)$, that generates the training data (say $\{z_i = (x_i, y_i)\}_{i=1}^{n}$, with $x_i$ the input and $y_i$ the class label belonging to one of the $K$ classes.), one can obtain the parameters $\theta^*$ corresponding to $G^*$ as follows,

$$\theta^* = \arg\min_\theta \mathbb{E}_{z \sim G^*}[L_\theta(z)] = \arg\min_\theta \int L_\theta(z)dG^*(z) \tag{9}$$

However, in reality we only have finite number of data points, and $G^*$ is of course unknown, and therefore the uncertainty in $\theta$ and consequently in the predictions of the (neural-network) model emerge.

Recall from previous Section that a DP prior, $\text{DP}(\alpha, F_0)$ serves as a distribution over the set of distributions, and additionally support of these distributions lie within the support of $F_0$ (Lemma 1. Correspondingly, we begin by assuming that $G^* \sim \text{DP}(\alpha, F_0)$, choosing $F_0$ such that it encapsulates the support (which we assume to be known) of the true-underlying distribution, and incorporates any other available information about $G^*$.

Next, based on the finite number of data-set, $\{z_i\}_{i=1}^n$, available (training set), we obtain the corresponding DP posterior, $\text{DP}(\alpha_n, \overline{F_n})$, where,

$$\alpha_n = \alpha + n, \; \overline{F}_n = \frac{n}{\alpha + n} F_n + \frac{\alpha}{\alpha + n} F_0 \tag{10}$$

where $F_n$ is the empirical distribution. After this, we sample a number of i.i.d. random measures, $P_j \sim \text{DP}(\alpha_n, \overline{F_n})$, $j = 1, ..., J$ posterior utilizing the finite stick breaking formulation formula for DP-posterior developed in previous section, Equation 8.

$$P_j(\cdot) \stackrel{d}{=} V_n \delta_{z_n}(\cdot) + \sum_{i=1}^{n-1} \left[ V_i \prod_{j=i+1}^{n} (1 - V_j) \right] \delta_{z_i}(\cdot) + \left[ \prod_{i=1}^{n} (1 - V_i) \right] P_0(\cdot) \tag{11}$$

where, $V_i \sim \text{Beta}(1, \alpha + i)$, and $P_0 \sim \text{DP}(\alpha, F_0)$, and $P_0$ is again sampled using the finite stick breaking formulation for DP prior (Equation 5), i.e.

$$P_0(\cdot) = \sum_{i=n}^{n+T} p_i \delta_{z_i}(\cdot) \tag{12}$$

where T is the level at which the infinite series is truncated, and the weights $p_i$ can be calculated as follows,

$$p_n = V_n, \; (p_i)_{i=n+1}^{n+T-1} = V_i \prod_{j=1}^{i-1} (1 - V_j), \; p_{n+T} = \prod_{i=n}^{n+T} (1 - V_i) \tag{13}$$

$$V_i \stackrel{iid}{\sim} \text{Beta}(1, \alpha), \; z_i \stackrel{iid}{\sim} F_0, \; i = n, ..., n+T \tag{14}$$

Substituting Equation 12 into 11 one can write $P_j$ as,

$$P_j(\cdot) = \sum_{i=1}^{n+T} p_i' \delta_{z_i}(\cdot) \tag{15}$$

where the first $n$ probability weights, $\{p_i'\}_{i=1}^n$ correspond to the training data points, calculated as in Equation 11, and the other weights, $\{p_i'\}_{i=n+1}^{n+T}$, correspond to those generated through DP prior (or base measure), derived as in Equations 13-14 (and multiplied by the multiplicant corresponding to $P_0(\cdot)$ in Equation 11)

For each $P_j$ we can obtain a neural-network model, $\theta^j$, i.e. a sample from the posterior over neural-network models using Equation 9. It's easy to see that the discrete representation for $P_j$ in Equation 15 simplifies Equation 9 to,

$$\theta^j = \arg\min_\theta \sum_{j=1}^{n+T} p_j L_\theta(z_j) \tag{16}$$

The complete algorithm is summarized in Algorithm 1. In order to compute variations in the predictions and confidence intervals, one can compute predictions for each of the sampled model.

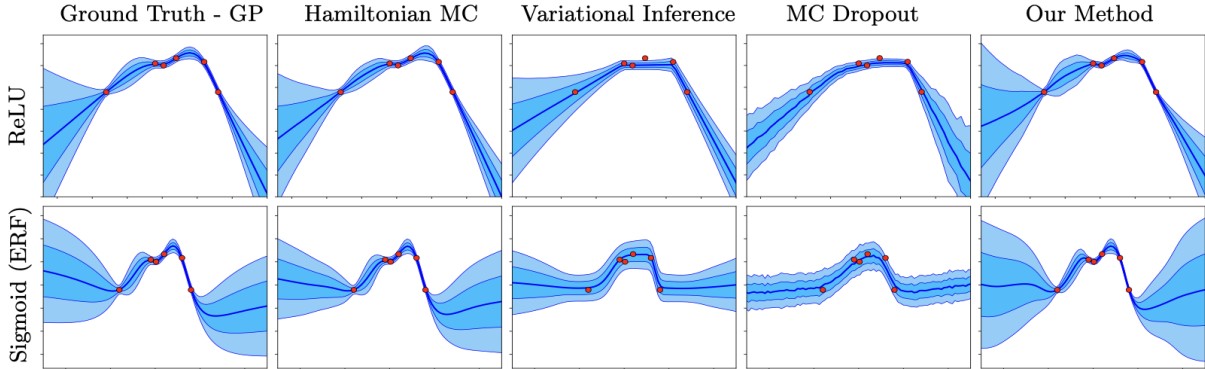

Figure 1: Comparison of DP-BNN with other Bayesian procedures for the toy data-set

Finally, recall from previous section, and observe again from Equation 15 that a random measure from DP posterior is supported not merely at original data-set points, but also those generated from the DP-prior (or correspondingly from the base measure $F_0$). When $\alpha \to 0$ one gets Bayesian Bootstrap (see previous section) version and gets random measures supported only at the data points. In other words, this DP based posterior-sampling scheme for neural-networks provides the flexibility to incorporate available prior information about true underlying distribution generating the training data, through the base measure of the DP prior, $F_0$.

---

**Algorithm 1** DP based scheme for sampling from posterior over neural-network models

---

1: **Input:** Observations $z_{1:n}$, DP-prior base measure $F_0$ and concentration parameter $\alpha$, DP prior truncation parameter $T$, Number of posterior samples $J$, neural-network architecture, neural-network loss $L_\theta(\cdot)$
2: **Output:** samples, $\{\theta^{(j)}\}_{j=1}^J$, from posterior over neural-network models
3: **Compute:** DP posterior, $\text{DP}(\alpha_n, \overline{F_n})$, with observations, $z_{1:n}$
4: **for** $j = 1, ..., J$ **do**
5:     sample a random measure, $P_j \sim \text{DP}(\alpha_n, \overline{F_n})$, Equation 15
6:     compute $\theta^{(j)} = \arg\min_\theta \int L_\theta(z) dP_j(z)$ using Equation 16
7: **end for**

---

Next, we show the performance of DP-BNNs on a toy dataset (generated using $z_i = sin(3x_i) + \epsilon, \epsilon \sim N(0,1)$.) in Fig. 1. We sample 5 data points and capture uncertainty in the predictions of a neural network with 10 hidden layers (for two different activation functions) plotting predictions corresponding to 10 different DP random measures (base measure is chosen as uniform distribution over $[0,1]$). The corresponding result is plotted in Fig.1. Note that, unlike other approximate Bayesian methods (MC dropout (Gal & Ghahramani, 2016) and Variational inference), the topology of the prediction intervals of our method (DP-BNN) matches with those for the ground truth, i.e. Gaussian processes (GPs), which correspond to the neural network model with infinitely many hidden units in the single layer (Ghahramani, 2015), and also with (computationally expensive) Hamiltonian MCMC (Neal et al., 2011) sampler for the posterior over the neural network models. The computational time for our method is similar to that of dropout in this case.

Next, we discuss the application of DP-BNNs for randomized value function algorithms for RL.

## 5 Dirichlet process based deep Thompson sampling for Markov decision processes

We begin by formalizing the setting of Markov decision processes (MDPs), and application of neural networks for generalization across continuous state-action spaces in reinforcement learning, and then introducing a DP-based deep Thompson sampling algorithm for MDPs.

### 5.1 Markov decision processes

MDPs model stochastic, discrete-time and finite action space control problems (Puterman, 1990). An MDP is a tuple $M = (\mathcal{X}, \mathcal{A}, R, P, \gamma)$ where $\mathcal{X}$ is the state space, $\mathcal{A}$ the action space, $R$ the reward function, $\gamma \in (0, 1)$ the discount factor and $P$ a stochastic kernel modeling the one-step Markovian dynamics ( $P(y \mid s, a)$ is the probability of transitioning to state $y$ by choosing action $a$ in state $s$ ). A stochastic policy $\pi$ maps each state to a distribution over actions $\pi(\cdot \mid s)$ and gives the probability $\pi(a \mid s)$ of choosing action $a$ in state $s$. The quality of a policy $\pi$ is assessed by the action-value function $Q^\pi$ defined as:

$$Q^\pi(s, a) := \mathbb{E}Z^\pi(s, a) = \mathbb{E}\left[\sum_{t=0}^{\infty} \gamma^t R\left(s_t, a_t\right)\right] \tag{17}$$

where $\mathbb{E}$ is the expectation over the distribution of the admissible trajectories ( $s_0, a_0, s_1, a_1, \dots$ ) obtained by executing the policy $\pi$ starting from $s_0 = s$ and $a_0 = a$, and therefore the quantity $Q^\pi(s, a)$ represents the expectation of random return $Z^\pi(s, a)$: $\gamma$-discounted cumulative reward collected by executing the policy $\pi$ starting from $s$ and $a$. Fundamental to reinforcement learning is the use of Bellman's equation, easily derived from Equation 17, to describe the value function:

$$Q^\pi(s, a) = \mathbb{E}R(s, a) + \gamma\mathbb{E}Q^\pi\left(s', a'\right). \tag{18}$$

In reinforcement learning we are typically interested in acting so as to maximize the return, and one utilizes the optimality equation corresponding to Equation 18,

$$Q^*(s, a) = \mathbb{E}R(s, a) + \gamma\mathbb{E}\max_{a' \in \mathcal{A}} Q^*\left(s', a'\right). \tag{19}$$

This equation has a unique fixed point $Q^*$, the optimal value function, corresponding to the set of optimal policies $\Pi^*$ ($\pi^*$ is optimal if $\mathbb{E}_{a \sim \pi^*} Q^*(s, a) = \max_a Q^*(s, a)$ ). Q-learning is the standard algorithm to compute optimal value functions in an iterative manner, with their updates given as follows,

$$Q_{t+1}\left(s_t, a_t\right) = Q_t\left(s_t, a_t\right) + \alpha_t\left(s_t, a_t\right)\left(r_t + \gamma\max_a Q_t\left(s_{t+1}, a\right) - Q_t\left(s_t, a_t\right)\right) \tag{20}$$

In this equation, $Q_t(s, a)$ gives the value of the action $a$ in state $s$ at time $t$. The learning rate $\alpha_t(s, a) \in [0, 1]$ ensures that the update averages over possible randomness in the rewards and transitions in order to converge in the limit to the optimal action value function. The discount factor $\gamma \in [0, 1)$ has two interpretations. First, it can be seen as a property of the problem that is to be solved, weighing immediate rewards more heavily than later rewards. Second, in infinite-horizon ($T \to \infty$) tasks, the discount factor makes sure that every action value is finite and therefore well defined. It has been proven that Q-learning reaches the optimal value function $Q^*$ with probability one in the limit under some mild conditions on the learning rates and exploration policy (Puterman, 1990).

### 5.2 Deep reinforcement learning

Deep Reinforcement Learning uses deep neural networks as function approximators for RL methods. Deep Q-Networks (DQN)(Mnih et al., 2015), Dueling architecture (Wang et al., 2016), Trust Region Policy optimization (Schulman et al., 2015) are examples of such algorithms. They frame the RL problem as the minimization of a loss function $L(\theta)$, where $\theta$ represents the parameters of the neural network. We will focus specifically on DQN. DQN (Mnih et al., 2015) uses a neural network as an approximator for the action-value function of the optimal policy $Q^\star(x, a)$. DQN's estimate of the optimal action-value function, $Q(x, a)$, is found by minimising the following loss (motivated by Q-learning update rule (Equation20)) with respect to the neural network parameters $\theta$,

$$L(\theta) = \mathbb{E}_{(x,a,r,y)\sim D}\left[\left(\left(r + \gamma \max_{b\in A} Q\left(y,b;\theta^-\right) - Q(x,a;\theta)\right)^2\right]\right. \tag{21}$$

where $D$ is a distribution over transitions $e = (x, a, r = R(x,a), y \sim P(\cdot \mid x, a))$ drawn from a replay buffer of previously observed transitions. Here $\theta^-$ represents the parameters of a fixed and separate target network which is updated ( $\theta^- \leftarrow \theta$ ) regularly to stabilize the learning. An $\epsilon$-greedy policy is used to pick actions greedily according to the action-value function $Q$ or, with probability $\epsilon$, a random action is taken.

### 5.3 DQN with DP based Bayesian neural networks

Note that although deep RL algorithms such as DQN have attained superhuman performance (Mnih et al., 2015), they can still fail at simple tasks that require efficient exploration (Osband et al., 2016). This is because they use statistically inefficient exploration schemes such as the one stated above for DQN, $\epsilon$-greedy, which approximate the value of an action by a single number, without accounting for the uncertainty in their estimates.

---

**Algorithm 2** DP based deep Thompson sampling for DQN

---

1: **Input:** neural network for Q value function, DP prior $(\mathrm{DP}(\alpha, F_0))$
2: Let B be the replay buffer storing training experiences of the agent
3: **for** every episode **do**
4:     Obtain initial state $s_0$ from environment
5:     Sample a Q neural-network model, $\theta^t$, from its posterior using Algorithm 1
6:     **for** time $t = 1, .., \tau$ **do**
7:         Select action $a_t = \arg\max_a Q_t(s_t, a; \theta^t)$
8:         Observe reward $r_t$, transition to $s_{t+1}$
9:         Add $(s_t, a_t, r_{t+1}, s_{t+1})$ to replay buffer $B$
10:     **end for**
11: **end for**

---

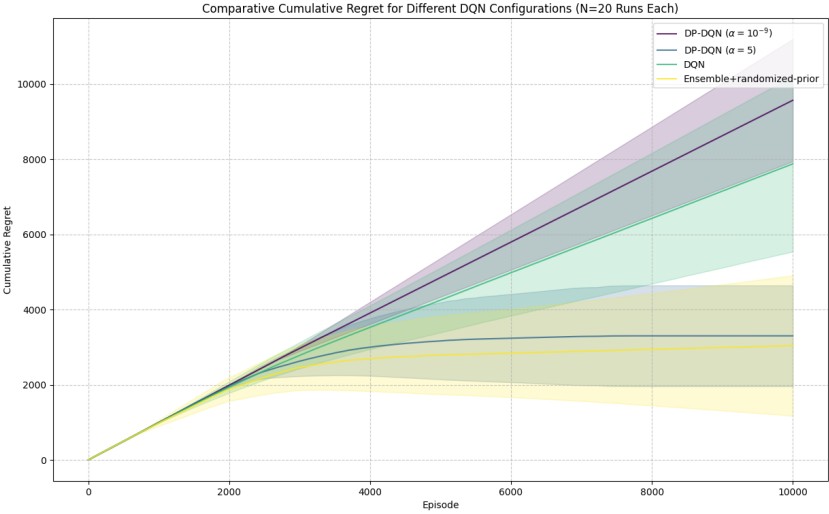

Figure 2: Performance of different RL algorithms in the deep-sea environment

DP-BNNs based deep Thompson sampling can be utilized to remedy this, because here one can utilize (Bayesian) uncertainty estimates (instead of a point estimate) over the action-value functions (each sample

from posterior over neural network models for action-value function would correspond to one estimate) for efficient exploration. We illustrate the algorithmic principle with an example for DQN, with corresponding pseudocode in Algorithm 2. Instead of plain deep neural network, we begin with a DP-BNN introduced in Section 4 for representing the action-value function in DP based deep DQN. Like in vanilla DQN, in each episode we sample the training data for the action-value neural network from the replay-buffer B, and sample a posterior for this neural network with this training data using algorithm 1 with the loss for neural network as DQN loss in Equation 21 . The sampled neural network model is then used to make decisions in the entire episode (Osband et al., 2013) as highlighted in algorithm 2 in the spirit of Posterior sampling reinforcement learning (Osband et al., 2013). Note from discussion on DP-BNNs from previous section that the corresponding DP-BNN based DQN algorithm would be based on two components of uncertainty – one based on randomization of value-functions through statistical (Bayesian) observed data-set (replay buffer), and second based on synthetically generated data from the base measures of the DP priors (see a discussion on the choice of these base measures in Sec B.2).

Next, we illustrate the performance of this DP-BNN based DQN (DP-DQN) algorithm on a challenging deep-exploration benchmark, $20 \times 20$ deep-sea environment of the Behavior-suite for RL (Osband et al., 2019a) in Fig. 2. Besides the exploration mechanism, all other factors for all the algorithms were same (e.g. size of replay buffer (128), neural network architecture (1 hidden layer with 20 neurons), and target network update frequency of 100 episodes, and cumulative regret averaged over 20 independent runs). Note that $\epsilon$-greedy based DQN suffer linear regret and only DP-DQN, and Ensemble (of 10 neural networks) + randomized prior network based DQN (Osband et al., 2019b) figures out the optimal policy by around episode number 3000 which is noteworthy because $\epsilon$-greedy style schemes have been shown to take around $2^{20}$ episodes to figure out the right policy for a $20 \times 20$ deep-sea, and essentially this also shows that DP-DQN is capable of deep-exploration. Also note that the Bayesian-Bootstrap version of DP-BNNs (i.e concentration parameter ($\alpha$) of the DP prior very close to zero) suffers linear regret which highlights the necessity of prior-data for deep-exploration, as was also discussed in (Osband et al., 2018), and shows how the DP based scheme introduced in this paper synthesizes the Bayesian Bootstrap and randomized prior in a neat Bayesian nonparametric manner.

Another observation is that, we used very simple base measures ( uniformly sampling one-hot encoding vectors of size $N^2$ for deep-sea environment, of size $N \times N$, for artificial states, uniformly random sampling from action space of the environment for artificial actions, and a normal distribution prior for the artificial rewards) for generating artificial history, and we believe that the performance can be improved further with a more careful choice of these base measures. Finally, note that the major computational overhead of DP-DQN above vanilla DQN lies in the generation of artificial history step, which is a small constant additional cost per episode. More concretely, on a standard laptop computer, for the above deep-sea runs, on average, 100 episodes of vanilla DQN took 4.5 seconds, DP DQN took 6.1 seconds, and Ensemble + randomized prior, which is slower because it utilizes an ensemble of neural networks for decision making, took 32.3 seconds.

## 6 Conclusions

We introduced a class of Bayesian neural networks (BNNs) which utilizes the Bayesian nonparametric priors on the space of distribution functions, i.e. Dirichlet Processes, to exploit the uncertainty about the data-generating distribution in order to capture the (Bayesian) uncertainty in the predictions, and showed their application for designing randomized value function based algorithms for RL. Crucially, DP based BNNs provide a principled Bayesian mechanism to capture uncertainty in value-function estimates through artificial-history/priors, which is essential for deep-exploration (Osband et al., 2018). Implementation of these randomized algorithms for other challenging RL environments, e.g. Montezuma's revenge, are planned as future work. Another interesting direction is theoretical analysis of the randomized value function algorithms based on DP based BNNs in the "stochastic-optimism" framework (Osband & Van Roy, 2017).

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

## A  Finite Stick breaking representation of Dirichlet Process priors

The finite stick-breaking representation of DP priors discussed in the main paper (Eqs.5-6) has been pivotal in the success of DP based Bayesian-nonparametric models. A major reason for this success is that such truncated representation is provably efficient (Ishwaran & James, 2001). Particularly, to quantify the accuracy loss owing to truncation consider the quantities, $T_N = (\sum_N^\infty q_i)^r$ and $U_N = \sum_N^\infty q_i^r$, where $N$ is the level at which the representation is truncated,

$$\mathbb{E}(T_N(r,a,b)) = \left(\frac{\alpha}{\alpha+r}\right)^{N-1}, \tag{22}$$

$$\mathbb{E}(U_N(r,a,b)) = \left(\frac{\alpha}{\alpha+r}\right)^{N-1} \frac{\Gamma(r)\Gamma(\alpha+1)}{\Gamma(\alpha+r)} \tag{23}$$

Notice that both expressions decay exponentially fast in $K$, and hence good accuracy is achieved for moderate $K$. Fig. 3 shows an application of this scheme to sample random measures from a DP prior, $\mathrm{DP}(\alpha, F_0)$ for

two different values of concentration parameter, $\alpha$. In order to give more intuition to appreciate the utility of DPs for nonparametric inference, We given an example on inference on a galaxy-dataset. We also used this (and some other) benchmarks to validate the performance of our StickBreaking module for DPPS.

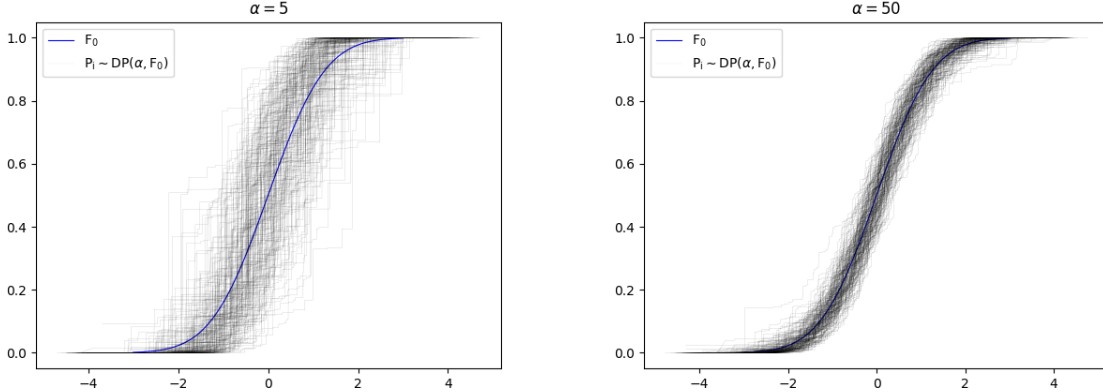

Figure 3: 200 random measures sampled from $\mathrm{DP}(\alpha, \mathrm{F}_0)$ where $\alpha = 5$ (left) and 50 (right), $\mathrm{F}_0 = N(0, 1)$

**DPs for galaxy data-set** We illustrate the application of Dirichlet processes for density estimation on a data set from the astronomy literature (Roeder, 1990). The measurements are velocities at which galaxies in the Corona-Borealis region are moving away from our galaxy. If the galaxies are clustered, the velocity density will be multimodal, with clusters corresponding to modes. This happens to be the case, and the multi-modal nature is evident in the CDF of the data in Figure 4 where the left and right regions of the CDF are almost flat, and most mass resides in the center. We estimate the CDF of this density using DP priors. Starting with a $\mathrm{DP}(\alpha = 2, \mathrm{F}_0 = \mathrm{N}(10, 1))$ DP prior, we obtain a DP posterior, and the spread of distributions sampled from the DP posterior can be seen as confidence-set of the CDF of density estimated through Dirichlet process.

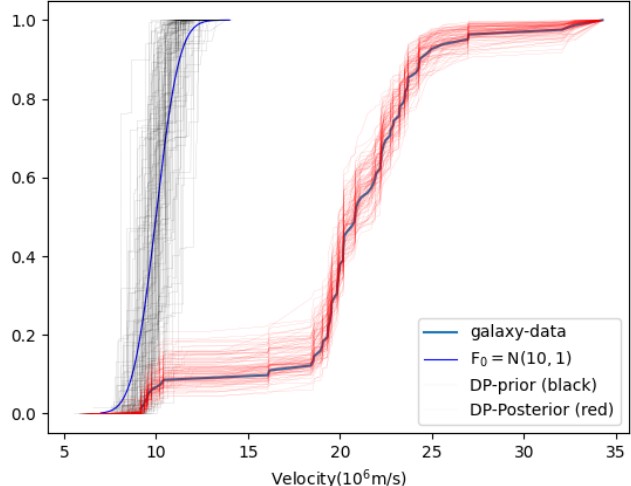

Figure 4: DP prior and DP posterior compared against original galaxy dataset distribution.

## B  Choice of hyperparameters in numerical experiments

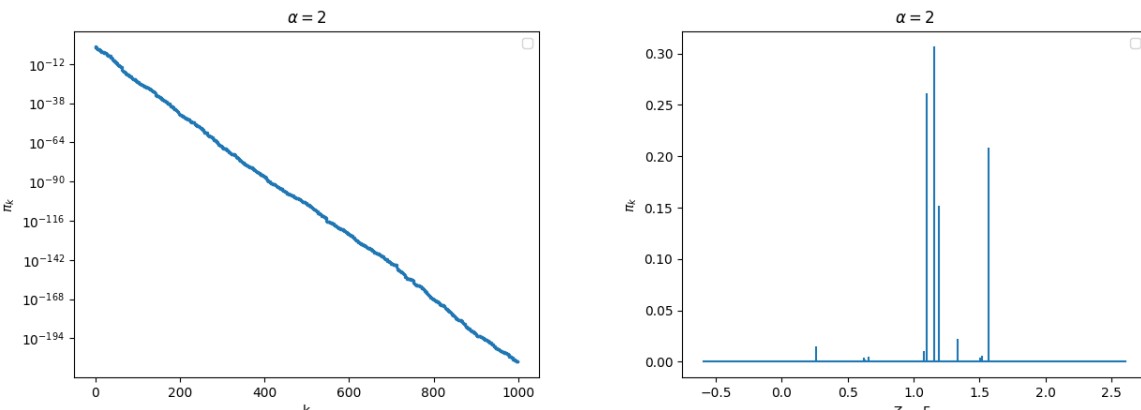

Figure 5: Plot of first 1000 stick-breaking probability measure weights, $\pi_k$, for $\mathrm{DP}(\alpha = 2, F_0)$ with k (left) and with $Z_k \sim F_0(= N(0, 1))$ (right)

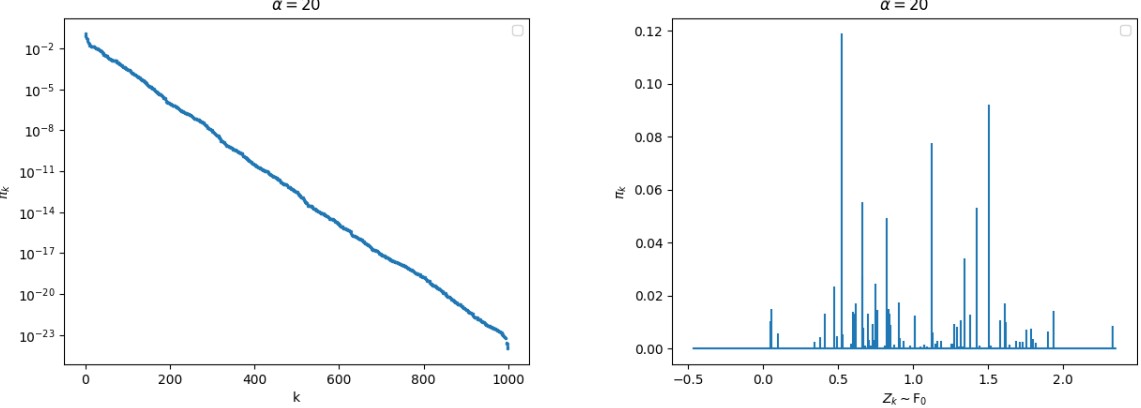

Figure 6: Plot of first 1000 stick-breaking probability measure weights, $\pi_k$, for $\mathrm{DP}(\alpha = 20, F_0)$ with k (left) and with $Z_k \sim F_0(= N(0, 1))$ (right)

### B.1  Choice of concentration and truncation parameters

Two hyperparameters in DP based BNNs are $\alpha$ (concentration parameter) and $k_t$ (i.e. truncation level) in the stick breaking representation of DP prior (not the posterior), $\mathrm{DP}(\alpha, F_0)$. We used $\alpha = 2$ and $k_t = 100$ in all the experiments. Note that the choice of $\alpha$ directly influences the choice of $k_t$. This is because the number of weights $q_i$ in the stick breaking representation, $\sum q_i \delta_{x_i}$, carrying significant probability mass increase with increase in $\alpha$ ($V_i \sim \mathrm{Beta}(1, \alpha)$), and for higher $\alpha$ one needs to increase $k_t$. For example, with $\alpha = 20$, we took $k_t = 300$, and we got similar results, with a slight increase in computation cost though. An easy way to determine $k_t$ is to plot the stick breaking weights and remove stick breaking weights that are below a certain threshold (we chose $10^{-10}$ randomly). This relationship between $\alpha$ and stick breaking probability weights, $q_i$, can be seen in a simple example of $\mathrm{DP}(\alpha, F_0)$ as shown in figs. 5 and 6. Whereas for lower value of $\alpha$ only few weights have significant mass, for higher $\alpha$ the weights are more evenly spread compared to lower $\alpha$ case.

### B.2 Choice of the base measures of the DP priors for the DP based DQN of algorithm 2

Note that, in the beginning episodes, there are no/very-less observations in the replay buffer, and the contribution of the (synthetic) data generated through DP prior would be significant in making decisions, and since the choices made initially can have significantly varying delayed consequences in RL, a proper choice of the DP prior (correspondingly the base measure of the DP prior) is crucial. A natural way to choose the base measure for the DP prior is to use product base measures with $F_0 = F_{0x}F_{0a}F_{0r}$. $F_{0x}$ and $F_{0r}$ would heavily hinge on the support of the MDP transition kernel $P(\cdot|x,a)$, and reward distribution $R$ respectively – One should choose $F_{0x}$ such that its support encapsulates the support of $P(y|x,a)$ and same holds for $F_{0r}$ wrt $R$. This is necessitated by the following lemma on the support of DPs,

**Lemma 1** (Support of DPs, see (Ghosal, 2010)). *In the weak topology, the support of $DP(\alpha, F_0)$ is characterized as all probability measures $P^\star$ whose supports are contained in that of $F_0$*

To put this into context consider an example: if $R$ is known to be $\sigma$-subgaussian then one should choose $F_{0r}$ to be a $\sigma_0$-subGaussian (e.g. $N(\mu, \sigma_0)$, $\sigma_0 \geq \sigma$, but not, for example, a Beta distribution because Beta has bounded support). The action space may be continuous for the MDP setting, and we should choose $F_{0a}$ based on the nature of the action-space, i.e. categorical distribution with $|\mathcal{A}|$ entries if $|\mathcal{A}|$ is discrete and finite, else choose $F_{0a}$ such that it encapsulates the bounds of the action-space (say bounded between 0 and 1 if on the real line, etc).

