# OpenReview forum: "Bayesian neural networks with Dirichlet-process priors on the data-generating-distribution for randomized value-functions based deep reinforcement learning"
_TMLR — Under review for TMLR_

### Review · Reviewer_8WJk · 2026-06-18

**Summary Of Contributions:**

The paper introduces a new family of Bayesian DNNs for capturing epistemic uncertainty, based in Dirichlet processes.
As a test case, the authors use RL and show that the new class of BDNN is capabale of enabling efficient information in a hard exploration task which requires effect reasoning about epistemic uncertainty.

*I'll preempt the review by stating that I'm from the area of RL, and reading this submission as a (deep) RL reader.*

**Strengths:**
1. The idea seems natural and elegant.
2. The significance seems very high - Bayesian approaches are notoriously good at in-distribution uncertainty estimation, and tend to fail in out of distribution detection. The experiments presented in this work make it clear that rather than a fail case, this is a success case of the method.
3. The results act as a very effective proof of concept for the method.
4. The presentation follows a structure I'm not used to, iterating background / novelty / background / novelty. I think this structure serves the submission very well, resulting in for the most part very clear and very well flowing presentation, that manages to encompass multiple areas (probability theory, DNNs, and RL) very well.

**Weaknesses:**

*Experiments:*
1. Figure 1: excellent, no notes.
2. Figure 2, and the deep sea results:
   1. The figure shows that the method is able to solve 20x20 Deep Sea. 20x20 is still a bit small for a real proof of concept for a functioning reliable epistemic uncertainty estimation / OOD detection method, in my opinion. Methods usually go until at least 50x50, which was the original benchmark (see [1]). Note that it's possible these results are actually for 50x50, however. bsuite ids the different experiments with an index. Each index corresponds to a different environment size, and deep-sea/20 corresponds to 50x50.
   2.  The figure also shows that BootDQN is unable to solve Deep Sea. This is not in line with previous results, again see [1]. I suspect the reason is the small-ish ensemble size 10. BootDQN is very sensitive to ensemble size. I would use the original baseline, which I believe was 20 or 30.
   3. The figure does not appear to include multiple seeds and statistical significance analysis.

*Clarity:*
As an RL reader, I find sections 2 and 3 hard to understand. The explanation is rather technical. For a technical explanation, I actually find it rather good, merely outside my area of expertize. However, I am missing a clearer conceptual explanation of the entire process.
Specifically, it's not clear to me how the method works in practice (eg Algorithm 1):
1. How is the DP posterior computed in practice? (line 3 in algorithm 1).
2. What does equation 15 amount to in practice?
3. Equation 16 seems computationaly intractable, fully minimizing a loss just to sample one model. Do I misunderstand? If not, how do we overcome that?
4. As a results of these questions, im not sure how to understand / assess the computational cost of the method.

*Related work:* I believe there is much more related work on Bayesian DNNs which could be discussed (which goes outside of my area of expertize), which I think is missing and should be discussed.

[1] Osband et al. "Behaviour suite for reinforcement learning." ICLR 2020.

**Audience:**

Yes

**Audience Explanation:**

The paper introduces a new family of Bayesian DNNs for capturing epistemic uncertainty, based in Dirichlet processes.
The method is able to achieve reliable out of distribution detection, as exemplified by the results on Deep Sea (to an almost sufficient extent, in my opinion, see weaknesses).

The significance is, in my opinion, high - Bayesian approaches are notoriously attractive from a theory perspective and good at in-distribution uncertainty estimation bu tend to fail in out of distribution detection, so this presents a significant step in this direction, from my familiarity with the area.

**Broader Impact Concerns:**

I don't believe a Broader Impact Statement is necessary.

**Claims And Evidence:**

Yes

**Claims Explanation:**

The submission makes a very clear set of claims in the abstract. I quote:
"
1. We introduce a new class of Bayesian Neural Networks (BNNs) which capture (Bayesian) uncertainty in predictions by exploiting the uncertainty about the underlying training-datageneration-distribution via treating it as a random variable distributed according to Bayesian
nonparametric priors on the space of distribution functions, i.e. Dirichlet Processes (DPs).
2. We show that these DP based BNNs provide a generalized Bayesian framework for designing randomized value-function based deep reinforcement learning (RL) algorithms.
3. Crucially, RL with DP-BNNs enables to introduce a "prior" mechanism in a principled Bayesian manner.
"

All three claims are supported by sections 3-4. Section 3 shows how this can be done in general for DNNs, and section 4 shows that this can be made to work in RL.

**Requested Changes:**

**Experiments:**
1. Please make sure that the results are already / please report results with Deep Sea 50 by 50. In fact, a stronger experiment in my opinion would be an experiment across growing problem sizes, see Figure 4 b in [1]. If this is feasible, I would add that. If it is not, could you explain how this relates to the computational cost of the method?
2. Please iterate on the bootdqn agent (implementation / just use the deepmind baseline available on line, ensemble size, etc.) to the point where it compares to the results of previous papers (or explain why there is a reason for it to fail in this domain, despite previous methods showin that it succeeds reliably). See [2] as another previous paper where bootdqn is able to solve deepsea within these budgets.
3. Please repeat the experiment in Figure 2 a sufficient number of times for a statistical significance analysis (or argue why it is not necessary).

**Clarity:**
1. Please include a more step-back, conceptual explanation of the method, specificaly section 3.
2. Please include an additional description of implementation details for the proposed BNNs, from a less-theoretical, more practictioner reader's perspective.
3. Please include a brief (if there's not much to discuss) discussion of computational complexity.

**Related work:**
Please include a dedicated related work section discussing uncertainty estimation methods (recent and classical), other approaches for bayesian DNNs, and contrasting them with the contributions of this work. The purpose is to give the reader a clear perspective of the contributions of this work in relation to other (recent or otherwise) works in the area.

**Minor comments:**
1. I think that posterior sampling can (and should) be introduced better. Referring to its theoretical guarentees, dominance in comparison to other methods (UCB based exploration, which is also theoretically motivated), etc. A better description of the possibilities and the difference between posterior over Q functions and posterior over models, and solving in the models, etc.
4. Section 2 in the middle of the first paragraph, there is a "It" (capital I) in the middle of the sentence.
5. The equations miss spacings (\quad s) between the different terms. For example, Equation 1, but almost all equations. Makes the reading much harder, because it's hard to seperate different equations within the same line in a glance. Whenever space allows, please include a \quad or so between each pair of equations in the same line.
7. There is an `Ofcourse` (instead of `Of course`) after Equation 6.
8. The brackets for $\gamma$ after 4.1 are flipped.
9. In Eq 17, I would suggest including the full depedence on the policy and the transition distribution in the equation, not just in the text. Eg $ E \Big[{\textstyle\sum\limits_{t=0}^{H-1}}
\gamma^{t}\mathcal{R}(s_t, a_t) \Big| {s_0 \sim \rho, s_{t+1} \sim P(s_t, a_t), a_t \sim \pi(s_t)} \Big]. $ There is the space for it, and it allows to actual write it such that everything is fully defined.
10. I would use "infinite horizon" instead of "non-episodic". In practice, even in infinite horizon domains we train the agent episodically, merely with bootstraps instead of terminals.
11. Please use latex's "ref{}" for equations, figures and Algorithms, so that they link.
12. I don't think there's a reason to use "Eq." instead of "Equation".
13. The symbols V_i and Q_i are used in Section 2 for one purpose other than value functions, and later used for value functions. These are very heavily used terms in RL (value / action value functions, as used later in the rest of the paper). Unless this choice was made specifically to connect to value functions in the later part of the paper (in which case, Im unable to understand the connection), to make it easier to read for a target audience of RL readers, please use different symbols. Similarly for small q.

[2] Oren et al. "Epistemic monte carlo tree search." ICLR 2025.

---

### Review · Reviewer_5NQy · 2026-06-21

**Summary Of Contributions:**

The paper places the Bayesian prior on the data-generating distribution rather than on the network weights. That distribution is modeled as a random measure with a Dirichlet Process prior, and posterior samples are drawn through the Stick-breaking and iterative DP constructions.
The main contribution is to show that this single construction unifies two mechanisms used in randomized value-function RL, data randomization and prior injection.
The paper demonstrates this as a DQN variant with Thompson sampling and reports that it solves the deep-sea environment while $\epsilon$-greedy based DQN and BootDQN incur linear regret. The clean unifying view of bootstrap and prior injection as one DP posterior is the key strength.

**Audience:**

Yes

**Audience Explanation:**

Researchers in Bayesian deep RL and exploration would find the framing worth reading. Casting data randomization and prior injection as two regimes of a single Dirichlet Process posterior is a clean unification of mechanisms the randomized value-function literature treats separately. The interest lies in this construction. The empirical results do not yet give that audience a result to rely on, but the conceptual contribution alone meets the audience criterion.

**Broader Impact Concerns:**

No broader impact concerns arise from the methodology or the benchmarks.

**Claims And Evidence:**

No

**Claims Explanation:**

While the conceptual claims are adequately supported at the derivation level, the central empirical claim remains problematic. Since this paper focuses primarily on an empirical contribution rather than a theoretical one, the core claims must be backed by rigorous statistical validation.
Specifically, three major weaknesses undermine this empirical claim.

First, the deep-exploration evaluation relies on a single run in a single environment without any random seeds or confidence intervals. This fails to establish reliable behavior, and additional evidence across a wider range of Behavior Suite tasks reported with multiple seeds is necessary.

Second, the crucial ablation study that would isolate the core hypothesis (the Bayesian Bootstrap variant without artificial history) is only parenthetically noted as 'not shown' on page 9, despite the necessity of this prior mechanism being the paper's central claim.

Last, the most critical baselines are missing, such as Bootstrapped DQN with randomized prior functions (Osband et al., 2018), which is known to solve the deep-sea environment, and Bayesian Deep Q-Networks (Azizzadenesheli et al., 2018). Without these comparisons, the evidence fails to distinguish the proposed method from existing prior-augmented or Bayesian deep RL approaches.

**Requested Changes:**

1. Broaden the experimental validation beyond a single deep-sea run. Since this paper contribution is empirical rather than theoretical, the claims need statistical support. Include multiple seeds with a measure of dispersion, more than one deep-sea size, and additional Behaviour Suite tasks beyond deep-sea, so that the deep-exploration claim is shown to hold across a range of benchmarks rather than on one favorable run.

2. Add the Bayesian Bootstrap ablation described as "not shown" on page 9 as a full result, since it is the evidence that isolates the central claim about the necessity of the prior mechanism.

3. Strengthen the related work on Bayesian neural networks for deep RL and add the corresponding baselines. At minimum, compare against bootstrapped DQN with randomized prior functions (Osband et al., 2018) and against Bayesian deep Q-networks (Azizzadenesheli et al., 2018), and discuss how the DP formulation relates to each. These are the comparisons that would let the contribution over existing methods be assessed.

4. Revise the abstract and the opening of the introduction so the argument runs from the limitation of weight-space priors to the proposed resolution, rather than stating the resolution first. Also, the paragraph beginning "To overcome these limitations" on page 2 largely restates the contributions list that follows it and could be compressed so the two do not duplicate each other.

5. Make the title more specific. The current title does not distinguish this work from other Bayesian neural network approaches to RL, and a title that names the data-generating-distribution or Dirichlet Process prior view would make the particular contribution legible relative to that body of work.

6. Typos
- The paragraph beginning "When" on pages 2 to 3 should read "when" rather than "When".
- The reference to "section" on page 2 should be capitalized as "Section".
- "delienated" on page 2 should read "delineated".
- "Ofcourse," on page 4 should read "Of course,".
- "syntheticall generated" on page 8 should read "synthetically generated".
- "artifical-history" should read "artificial-history" throughout.

Reference
- Azizzadenesheli, Kamyar, Emma Brunskill, and Animashree Anandkumar. "Efficient exploration through bayesian deep q-networks." 2018 Information Theory and Applications Workshop (ITA). IEEE, 2018.

---

### Review · Reviewer_ZVtZ · 2026-06-30

**Summary Of Contributions:**

The submission proposes Dirichlet-process Bayesian neural networks (DP-BNNs) as an alternative way to represent epistemic uncertainty. Instead of placing a prior directly over neural-network weights, the paper
i) places a Dirichlet process prior over the data-generating distribution,
ii) samples posterior random measures using stick-breaking or iterative DP posterior representations, and
iii) fits neural networks by minimizing a weighted empirical loss under each sampled measure.
This induces a distribution over trained neural-network predictors.

The paper then applies this construction to RL by proposing a DP-based Thompson-sampling variant of DQN, where a value-function network is sampled from the induced DP-BNN posterior.

**Audience:**

Yes

**Audience Explanation:**

The paper is at the intersection of several topics relevant to TMLR such as Bayesian deep learning, Bayesian nonparametrics, and uncertainty-aware RL. However, the current submission is not yet convincing and needs substantially stronger experiments, and comparisons against the most relevant modern baselines.

**Broader Impact Concerns:**

No concerns

**Claims And Evidence:**

No

**Claims Explanation:**

1. The empirical support is weak. Figure 1 uses only five data points, reports no quantitative metrics, no calibration scores, no coverage/error bars, no wall-clock measurements, and no details sufficient to reproduce the comparison. The statement that the DP-BNN uncertainty “matches” GP or HMC is only qualitative.

2. The claim that the computational time is similar to MC dropout is unclear and possibly misleading. MC dropout requires training a single network and using multiple stochastic forward passes at test time, whereas the proposed DP-BNN method appears to require training a separate neural network for each sampled DP posterior measure. Therefore, the method should have the training and storage costs of an ensemble method. The authors should distinguish training cost, inference cost, and wall-clock time. They should also compare against deep ensembles and Bootstrapped DQN-style ensembles, since those are more appropriate baselines than MC dropout.

3. The RL evidence is also insufficient for the claims made. The DeepSea result is only one environment size without seeds or confidence bands.

**Requested Changes:**

1. It is hard to believe the submission has been proofread:
- There are random spaces (or lack of space) in at least 5 places
- References alternates between, e.g., "Eq.1", "Eq. 1" and "Eq 15"
- Some brackets are opened and never closed, e.g., "(Lemma 1"
- MC dropout has a citation only the second time it is discussed
- \delta_x is introduced the second time it is used
- There are some incomplete sentences, e.g. "Entire family of NTTR", "all the algorithms were same"
- There are repeated spelling issues, e.g. “artifical,” “delienated,” and “syntheticall,”

2. The authors should add much stronger baselines:
- For the BNN component, comparisons should include scalable posterior-sampling or approximate-posterior methods such as SGLD, Laplace or KFAC-Laplace, SWAG/SWA-based posterior approximations, deep ensembles, and recent scalable BNN posterior samplers.
- For the RL component, the paper should compare against Bootstrapped DQN with randomized prior functions, not only BootDQN without priors. This is the direct baseline for the claimed contribution.